# Using Pharmacokinetic–Pharmacodynamic Modeling to Study the Main Active Substances of the Anticancer Effect in Mice from *Panax ginseng*–*Ophiopogon japonicus*

**DOI:** 10.3390/molecules29020334

**Published:** 2024-01-09

**Authors:** Lu Liu, Jing Lyu, Longfei Yang, Yan Gao, Bonian Zhao

**Affiliations:** 1Institute of Pharmaceutical Research, Shandong University of Traditional Chinese Medicine, Jinan 250355, China; 2021100041@sdutcm.edu.cn (L.L.); 60230015@sdutcm.edu.cn (J.L.); 60230001@sdutcm.edu.cn (L.Y.); 2Innovation Research Institute of Traditional Chinese Medicine, Shandong University of Traditional Chinese Medicine, Jinan 250355, China; 3Collaborative Innovation Center for Ecological Protection and High Quality Development of Characteristic Traditional Chinese Medicine in the Yellow River Basin, Jinan 250355, China; 4High Level Traditional Chinese Medicine Key Disciplines of the State Administration of Traditional Chinese Medicine, Pharmaceutics of Traditional Chinese Medicine, Jinan 250355, China

**Keywords:** *Panax ginseng*–*Ophiopogon japonicus*, UPLC–MS/MS, lung cancer, PK–PD modeling, main active substances

## Abstract

Ginseng Radix et Rhizoma Rubra (*Panax ginseng* C.A. Mey, Hongshen, in Chinese) and Ophiopogonis Radix (*Ophiopogon japonicus* (L.f) Ker-Gawl., Maidong, in Chinese) are traditional Chinese herbal pairs, which were clinically employed to enhance the immune system of cancer patients. This study employed the pharmacokinetic and pharmacodynamic (PK–PD) spectrum-effect association model to investigate the antitumor active substances of *P. ginseng* and *O. japonicus* (PG–OJ). The metabolic processes of 20 major bioactive components were analyzed using Ultra-Performance Liquid Chromatography–Mass Spectrometry/Mass Spectrometry (UPLC–MS/MS) in the lung tissue of tumor-bearing mice treated with PG–OJ. The ELISA method was employed to detect the levels of TGF-β1, TNF-α, and IFN-γ in the lung tissue of mice at various time points, and to analyze their changes after drug administration. The results showed that all components presented a multiple peaks absorption pattern within 0.083 to 24 h post-drug administration. The tumor inhibition rate of tumor and repair rate of IFN-γ, TNF-α, and TGF-β1 all increased, indicating a positive therapeutic effect of PG–OJ on A549 tumor-bearing mice. Finally, a PK–PD model based on the GBDT algorithm was developed for the first time to speculate that Methylophiopogonanone A, Methylophiopogonanone B, Ginsenoside Rb_1_, and Notoginsenoside R1 are the main active components in PG–OJ for lung cancer treatment.

## 1. Introduction

Among the various causes of cancer-related deaths, lung cancer globally holds a leading position in incidence and mortality rates, posing an inevitable threat to human health [1]. In the field of cancer treatment, a shift from relying on external interventions like surgery, chemotherapy, or radiotherapy to activating the human immune system is crucial [2]. Traditional Chinese medicine (TCM) has gained attention for its antitumor effects and ability to restore immune responses [3,4]. TCM’s multi-component composition often results in synergistic effects on various diseases, aligning with the principles of systemic immunity. Nowadays, research on these drugs has gradually gained attention in a modern context. *Panax ginseng*, known as Hongshen in Chinese, is a premium herb renowned for its capacity to invigorate qi and strengthen the spleen and lungs [5,6]. *Ophiopogon japonicus*, a long-used TCM plant for yin nourishment, bodily-fluid production, and lung disease treatment [7], has been combined with *P. ginseng* to develop a Shenmai Injection, a representative herbal formulation used clinically to enhance immune function in cancer patients [8]. Recent research focuses on the chemical composition, pharmacodynamics, and therapeutic mechanisms of PG–OJ [9,10,11,12]. Although the antitumor effect of PG–OJ has been preliminarily studied, a systematic investigation of its main active substances remains inadequate. Our previous research demonstrated that the PG–OJ extract (PG:OJ = 1:1) can effectively inhibit the proliferation and migration activity of A549 cells. Furthermore, we explored the prototype chemical composition of PG–OJ extract, discovering a higher content of rare saponins compared to either PG or OJ alone.

The complexity of PG–OJ components poses a significant challenge in elucidating their functional targets and underlying mechanisms. Therefore, a systematic evaluation of the multi-component effects is necessary to investigate the efficacy of PG–OJ. The PK–PD-spectrum effect correlation model integrates pharmacokinetic and pharmacodynamic blood/tissue concentration, time, and efficacy data, allowing for the interpretation of dynamic changes and synergistic effects within the body [13,14]. A Gradient Boosting Decision Tree (GBDT), a machine-learning algorithm suitable for “non-linear, small sample” prediction, is particularly suitable for analyzing effective components in traditional Chinese medicine [15,16]. The LightGBM model, derived from the GBDT algorithm, demonstrates an efficient capability in identifying crucial antitumor-effect-related peaks. With PK–PD correlation at its core, the LightGBM model can facilitate data analysis and relationship description between complex effective components in traditional Chinese medicine and biological effects. LightGBM can be employed for the correlation analysis of PK–PD studies, aiming to unravel the intricate relationships between complex traditional Chinese medicine’s active ingredients and their corresponding biological effects.

Our study aims to investigate the metabolic processes of major bioactive components in the lung tissue of tumor-bearing mice treated with PG–OJ. Additionally, we will examine PG–OJ’s pharmacological effects on tumor growth and cytokines in mouse lung tissues during antitumor treatment. By utilizing the GBDT algorithm to establish a PK–PD model, this study elucidates the pharmacological basis of PG–OJ for treating non-small-cell lung cancer (NSCLC) and provides a new research foundation for future investigations into PG–OJ’s pharmacological mechanisms.

## 2. Results

### 2.1. Method Validation

#### 2.1.1. Specificity

In our previous study, 69 compounds were identified in PG–OJ using UPLC–Q–Exactive–Orbitrap–MS for comprehensive chemical characterization. However, due to varying bioavailability, not all compounds could be thoroughly detected in lung tissue samples from tumor-bearing mice. In this study, based on UPLC–MS/MS analysis, we successfully identified and selected 20 prototype components for further pharmacokinetic studies. These prototype components have shown therapeutic potential in inhibiting cancer cells [17,18].

As shown in Figure 1, we compared the chromatograms of blank lung tissues of mice, lung tissue samples containing standard working fluid, and lung tissue samples collected 0.083 h post-administration of PG–OJ. There was no significant interference with endogenous substances within the analyte-retention time range, indicating that the established method was suitably specific.

#### 2.1.2. Linearity and Lower Limit of Quantification (LLOQ)

The calibration curves, correlation coefficients (R^2^), linear ranges, and lower limits of quantification (LLOQ) for analytes in lung tissue of mice were presented in Table 1. The correlation coefficients R^2^ are all greater than 0.9970, indicating a good linear relationship. The LLOQ for each analyte was determined by calculating the signal-to-noise ratio (S/N), which needed to be greater than 10. It is important to highlight that the lower limits of quantification for each component ranged from 0.005 to 50 ng/g, demonstrating the high sensitivity of the method.

#### 2.1.3. Precision and Accuracy

Table 2 summarizes the inter-day and intra-day accuracy and precision of 20 compounds at low, medium, and high concentrations. The precision range for all analytes ranges from 1.39% to 14.73%, and the accuracy range is from −14.86% to 13.01%. All values are within the acceptable range for biological-sample analysis, with a relative error (RE) < ±15% and a relative standard deviation (RSD) < 15%. The results indicate that the adopted method can be effectively applied to the quantification process of the 20 analytes, with high levels of accuracy and precision.

#### 2.1.4. Extraction Recovery and Matrix Effect

The extraction recovery rates and matrix effects for 20 analytes across three different QC concentration levels were presented (Table 3). All these values fell within an acceptable range (87.71% to 109.20% and 86.80% to 108.68%), suggesting that the extraction process is both stable and effective. Furthermore, no significant ion suppression was observed.

#### 2.1.5. Stability

Under the following conditions, the 20 analytes in lung tissue remain stable: short-term stability (4 °C for 12 h), freeze–thaw-cycle stability (−80 °C with three repeated freeze–thaw cycles), and long-term stability (−20 °C with 20 days of freeze–thaw). The respective RSD% values for short-term stability, freeze–thaw-cycle stability, and long-term stability are 2.11% to 18.90%, 0.62% to 19.09%, and 1.57% to 18.75%, respectively. The details are presented in Table 4.

### 2.2. Pharmacokinetics Experiment

This study employed Ultra-Performance Liquid Chromatography–Mass Spectrometry (UPLC–MS/MS) to examine the pharmacokinetic properties of 20 bioactive components in a cell-derived xenograft (CDX) model following oral administration of a 10.2 g/kg PG–OJ extract. The drug concentration–time curves and pharmacokinetic parameters of the 20 bioactive components were presented in Figure 2 and Table 5. The findings demonstrated that the drug concentrations of the 20 bioactive components exhibited a biphasic or multi-phasic phenomenon. This could be attributed to the release and elimination of drugs in the stomach, with only a small portion being absorbed into the bloodstream and distributed to the lung tissue, resulting in the initial peak. Subsequently, the drugs are released and absorbed persistently through the hepatic–intestinal and gastrointestinal circulations, generating a second peak. Moreover, the presence of multiple peaks could be associated with the hydrolytic effects of bacterial enzymes or the intermittent release of bile during the reabsorption of saponin components [19].

The majority of saponins are distributed quickly in tissues from plasma within 5 min of administration. Protopanaxatriol-type (PPT) ginsenosides (such as ginsenoside Rg_1_, Rg_2_, Re, Rh_1_, Rf, and notoginsenoside R1), Oleanolic-acid-type saponins (OA) ginsenoside Ro, and Ocotillol-type saponins (OT), pseudoginsenosides RT_5_ and 24(*R*)-pseudoginsenoside F11 show rapid absorption and reach their maximum concentration (C_max_) within 15 min of administration. However, Protopanaxadiol-type saponins (PPD) ginsenosides (such as Ginsenoside Rb_1_, Rb_2_, Rb_3_, Rc, Rd, and Rg_5_) reach their peak concentration after 12 h. This suggests that PPT ginsenosides enter the bloodstream more quickly, allowing for earlier absorption within the body. In lung tissue, the elimination of PPT, OA, OT ginsenosides, and two flavanone compounds occurs within 8 h, while the majority of PPD ginsenosides’ elimination is a slower process. Notably, although ginsenoside Rh_1_ has the longest half-life (6.9 h), it is still notably shorter than those of pseudoginsenosides RT_5_ (10.6 h) and PPD ginsenosides (7.41–66.6 h, excluding ginsenoside Rg_5_). The difference in elimination characteristics between PPT and PPD ginsenosides could be associated with differences in plasma protein binding. Alternatively, it could be due to the fact that only PPT ginsenosides are able to be transported by organic anion transporting polypeptide 2B1 (OATP2B1), despite all ginsenosides binding to rat organic anion transporting polypeptide 2B23 (OATP2B23) [20].

In the PG–OJ extract, the AUC_(0–∞)_ of PPD ginsenosides Rg_5_, Rb_1_, Rd, and PPT ginsenosides F2 and Rg_2_ were significantly higher than those of other components, suggesting that these five components have higher exposure levels, which is associated with their increased content in the PG–OJ extract. In lung tissue, the component with the highest exposure level is the rare ginsenoside Rg_5_, which is synthesized during the steaming process of ginseng and has been proven to possess substantial potential bioactivity as a broad-spectrum anticancer and anti-inflammatory drug [21]. In contrast, the tissue exposure levels of PPD ginsenosides F1, Rh_1,_ and NotoG-R1 are extremely low, which may be attributed to Phase II enzymes metabolizing triterpene ginsenosides into water-soluble metabolites that can be excreted by the kidneys [22].

### 2.3. Pharmacodynamic Experiment

As illustrated in Figure 3 and Table 6, the administration of PG–OJ in the nude-mouse CDX model led to a rapid onset of action for the three cytokines IFN-γ, TNF-α, and TGF-β1 within 15 min. These cytokines reached their lowest concentrations at 8 h and remained at a relatively stable level until 24 h. Significantly, at 30 min post-PG–OJ intervention, the TGF-β1 concentration in the experimental group was notably higher than that in the model group. This finding suggests that during the early stages of tumor development, PG–OJ may upregulate TGF-β1 to facilitate apoptosis and inhibit tumor-cell proliferation. In addition, the tumor-growth-inhibition rate demonstrates a significant therapeutic effect after PG–OJ intervention (*p* < 0.001). 

### 2.4. Related Analysis

#### 2.4.1. Comprehensive Weighted Scoring of Efficacy Indicators

Information entropy serves as a probability measure to characterize the level of data discreteness. The smaller the entropy value, the greater the dispersion of the data. The entropy-weighted method can objectively determine the weights of each indicator and reflect various information objectively and accurately through weighted calculation. In this study, the comprehensive evaluation index of the efficacy of three cytokines (E%) was calculated using the entropy-weighted method, and the results are presented in Appendix A.

#### 2.4.2. Feature-Sorting Algorithm Based on LightGBM Model

This study proposes a feature-ranking approach based on the LightGBM algorithm, which reorders the original features to assign corresponding shadow features to each variable. The significance of each variable is assessed by comparing the permutation accuracy of true samples with the best shadow feature samples [23]. Taking into account variables that are meaningful to the model, the concentration ranges of 20 compounds were designated as the feature sequence, and the comprehensive effect index S for each sample served as the parent sequence. Based on the Kennard-Stone algorithm, a classic method, we randomly selected 80% of the samples as the training set and the remaining 20% as the testing set [24]. Additionally, to prevent model overfitting, we limited the maximum tree depth to five and set a minimum sample number for each leaf node. As shown in Figure 4A,C, the results from both the training and testing sets confirmed that the top-five feature variables (including Methylophiopogonanone A, Methylophiopogonanone B, Ginsenoside Rb_1_, Rh_1_, and Notoginsenoside R1) account for over 50% of the total SHapley Additive ex-Planations (SHAP) values, suggesting a significant impact on the model’s regression prediction. In Figure 4B,D, red points in the positive-influence graph represent features with a positive impact on the prediction result, while blue points indicate the opposite. Ginsenoside Rh_1_, Rf, Rb_3_, Rg_5_, and Rg_1_ exhibit a negative correlation with the comprehensive-drug-effect-index sequence. Finally, based on the positive correlation between the prototype components and S, we inferred that the active substance basis of PG–OJ against lung cancer includes Methylophiopogonanone A, Methylophiopogonanone B, Ginsenoside Rb_1_, and Notoginsenoside R1.

## 3. Discussion

Pharmacokinetics play a crucial role in establishing concentration–activity/toxicity relationships, facilitating target identification in traditional Chinese medicine, and the discovery of new drugs [25]. In pharmacokinetic studies, investigators typically explore one or more active ingredients in herbs, examining their tissue distribution and concentration–activity profiles. However, some compounds may be undetectable or contribute minimally to blood exposure, rendering them irrelevant to clinical outcomes. To address this limitation, machine-learning and activity-weighted approaches are frequently employed to investigate the exposure–efficacy/toxicity relationships of key efficacy component groups in target organs. This approach enables the explanation of the drug mechanism of action and identification of core components responsible for efficacy.

This study, based on the verification of the cytokine levels and tumor growth between MG and CG, successfully establishes the CDX nude-mouse model. Our previous research has demonstrated that PG–OJ extracts exhibit excellent anti-lung-cancer therapeutic effects at a four-fold-clinical-equivalent dose. In this study, we further validate the efficacy of this dose and explore its pharmacokinetic characteristics. The results indicate that oral administration of PG–OJ extract significantly inhibits tumor growth in tumor-bearing mice, with an average inhibition rate of 55.73%. These differences may be closely related to the strong absorption and slow elimination of some key active components in PG–OJ in vivo. In addition, PG–OJ extract can initially stimulate the secretion of IFN-γ, thereby initiating the antitumor immune process. Activated immune cells, such as T cells, then release other cytokines, including TNF-α and TGF-β1. TNF-α can promote tumor-cell apoptosis, while TGF-β1 plays a role in suppressing immune responses in the tumor microenvironment [26,27]. These three factors jointly participate in regulating the tumor immune microenvironment, affecting the development and treatment effect of lung cancer. In tumor-bearing mice, the dynamic changes in IFN-γ and TGF-β1 may be related to the expression-like tumor-cell levels, immune-cell functions, and other factors in the tumor microenvironment [28,29].

Traditional Chinese medicine compatibility may significantly affect the pharmacokinetic (PK) parameters and blood concentrations of drugs. Employing a sensitive UPLC–MS/MS method to analyze the time-varying curves of 20 compounds in the PG–OJ extract, with a focus on the lung as the target organ, offers a valuable approach to understanding the pharmacokinetic parameters and blood concentrations of drugs. In the PK studies, significant differences were detected in PK parameters between PPT-type and PPD-type saponins. These results align with previous findings, indicating higher exposure levels of PPD-type ginsenosides and slower elimination compared to PPT-type ginsenosides. Notably, rare ginsenosides such as Rg_5_ (i.e., the product of dehydroxylation and carbon dehydration at position C-20 of Rg_3_) exhibit higher tissue exposure levels. The study by Zhang et al. revealed that the number of sugar moieties in ginsenosides is an important consideration in their anticancer biological activity, with the level of anticancer biological activity decreasing proportionally with the number of sugar moieties [30]. 

We found that the concentrations of 20 chemicals in lung tissues were consistent with the changes in the improvement rates of the three cytokines. Notably, the improvement rates of IFN-γ, TNF-α, and TGF-β1 reached a second peak at 3 h after drug administration, which was later than the second peak of drug concentration (2.5 h after drug administration), indicating a delayed pharmacodynamic effect. Further analysis may require more in-depth research on pharmacokinetic parameters such as drug concentration and half-life for both the model and treatment groups. LightGBM-algorithm-based PK-PD spectral efficacy correlation analysis has revealed that four active ingredients, including Methylophiopogonanone A, Methylophiopogonanone B, Ginsenoside Rb_1_, and Notoginsenoside R1, have high characteristic-importance rankings and are positively correlated with the comprehensive efficacy index. Methylophiopogonanone A emerged as a crucial chemical indicator for the quality control in Ophiopogonis Radix-related herbal formulations. Existing studies have demonstrated that high concentrations of flavanone components may play a vital role in the treatment of lung cancer [31,32]. Ginsenoside Rb_1_, another compound, enhances the immune response by inducing antigen-presenting cells that secrete TNF-α and T cells that secrete interferon γ and IL-10 [33]. Additionally, ginsenoside Rb_1_ also promotes the production of immunoglobulins (such as IgA, IgG1, and IgG2) and enhances virus-triggered interferon γ expression [34]. Notoginseoside R1 exhibits anticancer activity by inhibiting TNF-α and suppressing cancer cell proliferation [35]. 

In conclusion, we suggest that Methylophiopogonanone A, Methylophiopogonanone B, Ginsenoside Rb_1_, and Notoginsenoside R1 may be the important pharmacological basis of the antitumor immune response in PG–OJ extracts. Utilizing PK–PD correlation-analysis methods can provide more accurate and reliable results in the mining of active ingredients within traditional Chinese medicine, thereby strongly supporting the development of promising new drugs. This study only focused on the antitumor effect of PG–OJ and did not compare it with other drugs. Future studies can further expand the sample size and explore the combination of PG–OJ with other drugs.

## 4. Materials and Methods

### 4.1. Materials and Chemicals

*P. ginseng* samples were purchased from Shennong Pharmacy (China, batch number: C21110403), while the *O. japonicus* was obtained from the *Ophiopogon japonicus* (Thunb.) Ker Gawl. plantation base in Sichuan, Shandong province (China). The human lung adenocarcinoma cell-line A549 cells were obtained from the Chinese Academy of Sciences (Shanghai, China). Ginsenoside Ro (CAS: 34367-04-9; Lot number: WKQ19012811; purity ≥ 98%), Rh_1_(CAS: 63223-86-9; Lot number: WKQ19013001; purity ≥ 98%), F1 (CAS: 53963-43-2; Lot number: WKQ19012509; purity ≥ 98%), F2 (CAS: 62025-49-4; Lot number: WKQ19012808; purity ≥ 98%), Rg_2_ (CAS: 52286-74-5; Lot number: WKQ18042804; purity ≥ 98%), Rg_3_ (CAS: 14197-60-5; Lot number: WKQ19012410; purity ≥ 98%), Pseudoginsenoside RT_5_ (CAS:98474-78-3; Lot number: WKQ19012409; purity ≥ 98%) and Pseudoginsenoside F11 (CAS: 69884-00-0; Lot number: WKQ18031309; purity ≥ 98%) were purchased from Weikeqi Biotechnology Co., Ltd. (Chengdu, China), whereas Rd (CAS: 52705-93-8; Lot number: PS0010161; purity ≥ 98%), Rb_2_ (CAS: 11021-13-9; Lot number: PS020544; purity ≥ 98%), Rc (CAS: 111021-14-0; Lot number: PS020852; purity ≥ 98%), Rf (CAS: 52286-58-5; Lot number: PS0010161; purity ≥ 98%), Rb_3_ (CAS: 68406-26-8; Lot number: PS020512; purity ≥ 98%), Rb_1_ (CAS: 41753-43-9; Lot number: PS011946; purity ≥ 98%) and Rg_5_ (CAS: 186763-78-0; Lot number: PS230911-02; purity ≥ 98%) were purchased from Chengdu Pusi Biotechnology Co., Ltd. (Chengdu, China). Re (CAS: 52286-59-6; Lot number: 110754-200421; purity ≥ 98%), Rg_1_ (CAS: 22427-39-0; Lot number: 110703-200424; purity ≥ 98%) and Notoginsenoside R1 (CAS: 80418-24-2; Lot number: 110745-201921; purity ≥ 98%) were purchased from China national institutes for drug control. (Beijing, China). Methanol, acetonitrile and formic acid (LC-MS grade) were purchased from Thermo company (Swedesboro, NJ, USA).

### 4.2. Preparation of Sample Solutions

A total of 120 g of dried powder from RG and OJ were accurately weighed into round-bottomed flasks, and then 70% ethanol (1:10, *w*/*v*) was added, followed by heating and reflux extraction for 1.5 h. The extract was filtered and recovered, and then dried to powder in a freeze dryer. Finally, the freeze-dried powder was dissolved in 0.3% sodium carboxymethyl cellulose water solution, resulting in a final concentration of 1.0 g·mL^−1^ (based on crude drug content). 

### 4.3. Animal Treatments and Tissue-Sample Collection

Healthy BALB/c nude male mice (weight: 18–21 g) were purchased from Beijing Huafukang Biotechnology Co., Ltd. (Beijing, China, License Number: SCXK (Jing) 2021-0006). After 3 days of acclimatization, the mice were randomly divided into the control group (CG), model group (MG), and the combined-administration group of *P. ginseng* and *O. japonicus* (HM1-HM13) (6.8 g/kg, four-fold clinical-equivalent doses). Subsequently, except for the CG, 0.2 mL (approximately 2 × 10^7^ cells) of A549 cell suspension was aspirated with a 1 mL syringe and injected into the right axillary fossa of the model group and drug-administration-group mice. The control-group mice were injected with an equal volume of PBS at the same location. The tumor diameter was measured daily using a caliper. The model was successful when the tumor diameter > 3 mm. The combined-administration group of *P. ginseng* and *O. japonicus* was given the corresponding drug orally, once a day, with 0.2 mL/mouse each time. The control group and the model group mice were given the same volume of distilled water, continuously orally for 21 days. During this period, the body weight and tumor volume of the mice were recorded. At the end of the last administration, the mice were weighed 0.083, 0.167, 0.25, 0.5, 0.75, 1, 2, 3, 4, 6, 8, 12, 24 h later. After injecting 0.1% pentobarbital sodium, blood was collected from the eye-socket vein, the mice were decapitated, and the tumor and lung tissues were carefully removed. 

After homogenizing 50 mg of right-lung-tissue sample, 2 mL of acetonitrile was added to precipitate proteins. The mixture was oscillated for 2 min and then centrifuged for 10 min (12,000 rpm, 4 °C). The supernatant from the new centrifuge tube was collected and dried with nitrogen at room temperature. Finally, the supernatant was redissolved in 50 μL of acetonitrile solution containing 0.1% formic acid. After oscillating for 1 min, another centrifugation step was performed, and the resulting supernatant was used for analysis.

### 4.4. UHPLC—Orbitrap MS Conditions

Samples were analyzed on the Vanquish UPLC System (Thermo, Waltham, MA, USA). The Agilent C18 column (4.6 × 150 mm, 4 μm, PN:693970-902T) was selected. The mobile phase was composed of 0.1% formic acid aqueous solution (A) and 0.1% formic acid acetonitrile solution (B). The procedure of gradient elution was as follows: 0–10 min, 95–75% A; 10–25 min, 75–45% A; 25–40 min, 45–30% A; 40–50 min, 30–5% A; 55–55.1 min, 5–95% A; 55.1–60 min, 95% A. The column temperature was set to 30 °C with the flow rate of 0.3 mL/min and the injection volume of 3.0 μL. The mass-spectrometry detection with negative-ion mode was carried out using an Orbitrap Exploris 120 Mass Spectrometer (Thermo, Waltham, MA, USA) and Electrospray ionization (ESI). The sheath-gas-flow rate was 30 L/min and the auxiliary-gas-flow rate was 10 L/min. The capillary temperature was 325 °C, and the resolution was 120,000 in full scan mode with the mass scan range of 80~1500 *m*/*z*. The secondary cracking was performed with HCD at a collision voltage of 30 eV, and unnecessary MS/MS information was removed using the dynamic exclusion method. Quantitative analysis was performed using the selective-ion-detection (SIM) mode (Appendix A). The total-ion chromatogram and chemical structures of the 20 compounds are shown in Appendix A.

### 4.5. Preparation of Quality-Control (QC) Samples and Method Validation

Twenty standard substances and internal standards (IS) were precisely weighed and separately dissolved in 70% methanol, followed by mixing and dilution to form a series of standard working solutions. The series of standard working solutions were then mixed with normal lung tissue to prepare QC (low, medium, and high concentration) samples, with an end concentration of 20 µg/mL for the internal standard. Appendix A shows the concentrations of each standard working solution and QC sample. According to the FDA Guidance for Industry on Bioanalytical Methods, the selectivity, matrix effect, extraction recovery, standard curve, accuracy, precision, stability, residual effect, and dilution effect of the HPLC–MS/MS method were all validated [36].

### 4.6. Data Analysis of the Pharmacokinetic Study

The Phoenix 8.1 software was used to analyze pharmacokinetic parameters, including the maximum concentration (C_max_), time to reach C_max_ (T_max_), elimination half-life (T_1/2_), mean residence time (MRT), and area under the concentration–time curve (AUC).

### 4.7. Pharmacodynamic Experiment

The tumor-inhibition rate for each group was calculated based on the body weight and tumor-tissue-sample weights of the mice. The left-lung-tissue samples from each group were homogenized in a PBS solution containing 1% PMSF (five times the weight of the lung tissue) and then collected after high-speed centrifugation. The concentrations of TNF-α, IFN-γ, and TGF-β1were measured using an ELISA kit according to the instructions of the kit, and the cell-factor repair rate was calculated. A curve is plotted showing the changes in tumor-inhibition rate and cytokine repair rate following drug intervention over time. All analyses were performed using GraphPad Prism Software 8.1. The cytokine repair rate (CR) and tumor-growth-inhibition rate (IR) were calculated as follows: CR (%) = (average cytokines levels of model group − average cytokines levels of treatment group)/(average cytokines levels of model group − average cytokines levels of control group) × 100%. IR (%) = 1 − (average tumor weight of treatment group/average tumor weight of model group) × 100%.

### 4.8. PK–PD-Correlation Analysis

#### 4.8.1. The Comprehensive Weight Method for Efficacy-Indicators Assessment

In this study, the comprehensive score S was calculated using the weight method, which covered CR. First, after processing the raw data using the Min–Max normalization method, the scaling of different indicators could be unified. Then, the dimensionless values {Y_ij_} were taken to calculate the characteristic weight value {P_ij_} as Equation (1). Subsequently, the information entropy value of parameter {E_j_} could be expressed as Equation (2). The comprehensive weight method used parameter entropy to calculate the weight of each indicator value {W_j_}, as shown in Equation (3). Finally, the comprehensive score {S} for the efficacy indicators of each sample was obtained as Equation (4).
(1)$$P_{\mathrm{ij}}=\frac{Y_{ij}}{\sum_{i=0}^{n}Y_{ij}}$$
(2)$$E_{j}=-k\sum_{i=1}^{m}P_{ij}\ln(P_{ij})\;k=1/\ln(m)>0\;\mathrm{and}\;E_{j}\geq 0$$


(3)
$$W_{j}=\frac{1-E_{j}}{\sum_{j=1}^{n}\left(1-E_{j}\right)}\;j=1,\;2,\;3,\;4,\;5$$



(4)
$$S=\sum_{j=1}^{n}W_{j}\times P_{\mathrm{ij}}$$


#### 4.8.2. Feature-Ranking Based on LightGBM Model

LightGBM is an algorithm based on ensemble learning that employs decision tree models to rank the importance of feature variables. By employing a gradient-based feature-selection method, it can effectively identify features with a strong influence on the target variable, thereby mitigating overfitting risks and enhancing the model’s generalization ability in small-sample scenarios. Therefore, it can be considered as a simple and effective comprehensive evaluation method of the spectrum–effect relationship. In LightGBM algorithm, the positive- and negative-force graph represents the degree of impact of features on the prediction results. SHAP value is a numerical method used to explain the prediction results of the model. It measures the contribution of each feature to the model’s prediction outcomes. The SHAP value ranges from −1 to 1, with values closer to 1 indicating a greater impact of the feature on the prediction result, and values closer to −1 indicating a smaller impact. In this study, a data matrix was constructed using the quantitative measurements of 20 component concentrations at different time points after administration as independent variables and the composite pharmacodynamic score S as the dependent variable. Z-score standardization preceded data matrix analysis, and Python 3.10 software was used to calculate the SHAP values of each component.

## 5. Conclusions

We developed a specific and sensitive LC–MS/MS method for concurrent quantification of 18 saponins and two flavanones in biological samples. Ginsenoside Rg_5_, a unique component of PG–OJ decoction, exhibits significant chemopreventive effects on cancer, suggesting it is a promising prospect. This method was effectively employed to study the pharmacokinetic properties of compounds in lung cancer mice by way of continuous gavage with PG–OJ extract for a period of twenty-one days. The improvement rates of TNF-α, TGF-β1, and IFN-γ indicate that PG–OJ can effectively enhance the antitumor immune effect. Based on the LightGBM algorithm, the PK–PD-pattern analysis revealed the importance ranking of 20 key components in lung cancer mice, and identified four key pharmacodynamic-substance-component groups (Methylophiopogonanone A, Methylophiopogonanone B, Ginsenoside Rb_1_, and Notoginsenoside R1). This research strategy aims to propose a practical and precise method, providing a scientific foundation for uncovering the key active ingredients of medicinal plants.

## Figures and Tables

**Figure 1 molecules-29-00334-f001:**
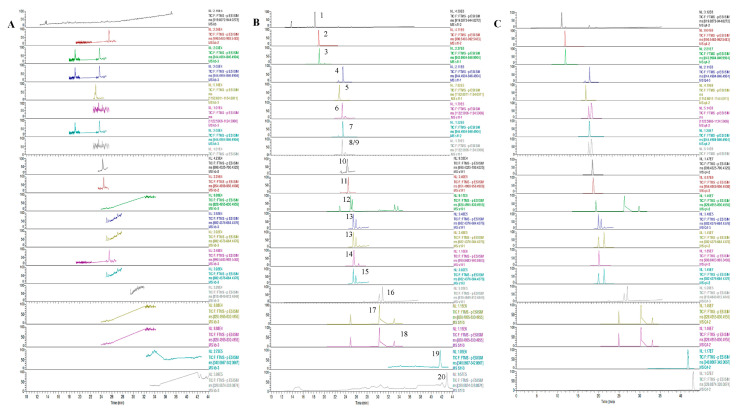
Extracted ion chromatogram of the 20 compounds. (**A**) Blank lung tissue samples; (**B**) lung tissue samples collected at 0.083 h after administration of PG–OJ; (**C**) blank lung tissue spiked with analytes. (1: Notoginsenoside R1, 2: Ginsenoside Re, 3: Ginsenoside Rg_1_, 4: 24(*R*)-pseudoginsenoside F11, 5: Ginsenoside Rb_1_, 6: Ginsenoside Rc, 7: Ginsenoside Rf, 8: Ginsenoside Rb_2_, 9: Ginsenoside Rb_3_, 10: Pseudoginsenoside RT_5_, 11: Ginsenoside Ro, 12: Ginsenoside Rg_2_, 13: Ginsenoside F1, 14: Ginsenoside Rd, 15: Ginsenoside Rh_1_, 16: Ginsenoside Rg_5_, 17: Ginsenoside Rg_3_, 18: Ginsenoside F2, 19: Methylophiopogon flavanone A, 20: Methylophiopogon flavanone B).

**Figure 2 molecules-29-00334-f002:**
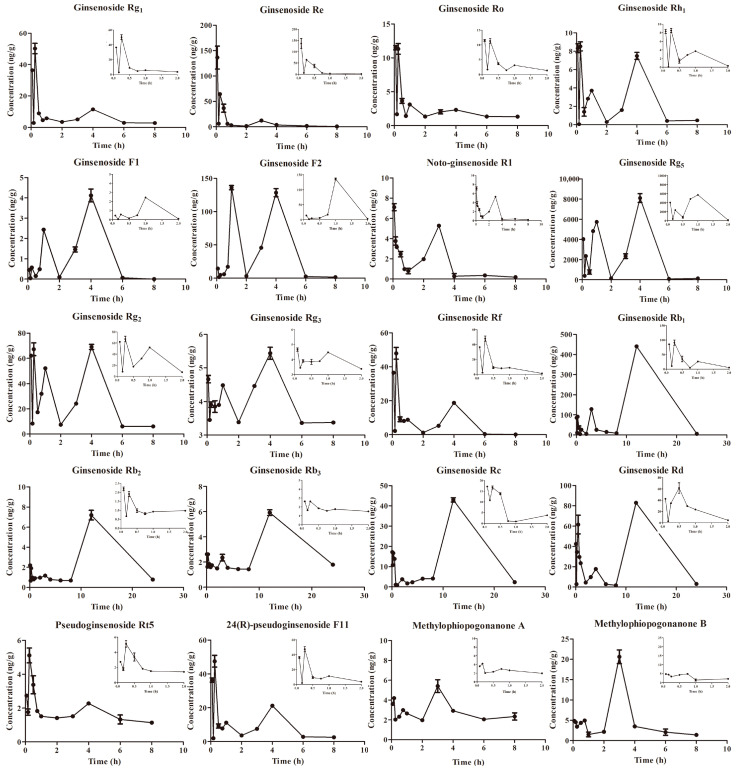
The profiles of mean lung tissue concentration; time of twenty target constituents after oral administration of PG–OJ extract to tumor-bearing mice. The main figure displays the average content changes of components in the lung tissue of six mice in the PG–OJ group within 24 h, while the secondary figure shows the average content changes within 2 h. The error lines represent the mean ± SD.

**Figure 3 molecules-29-00334-f003:**
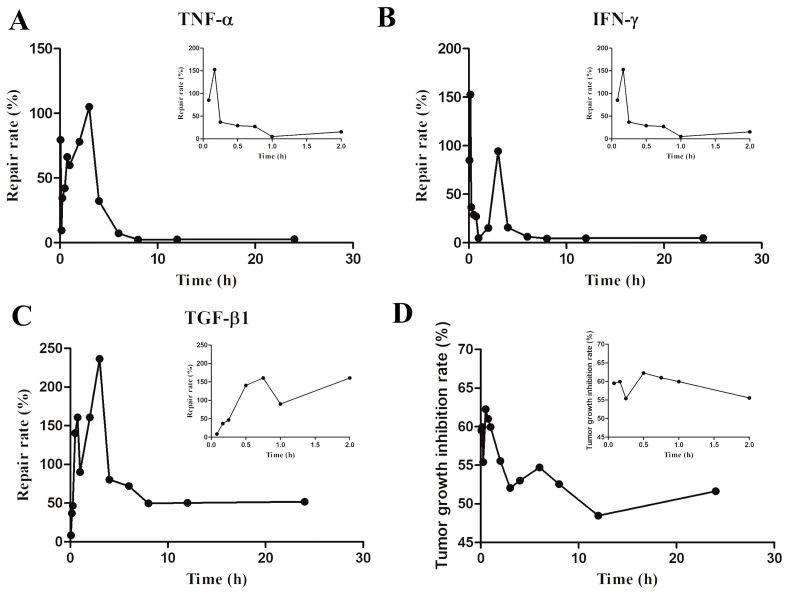
Pharmacodynamic indicator for changes in tumor-bearing mice after oral administration of PG–OJ extract at different time points. Main (**A**–**C**) present the average repair-rate changes of three cytokines (TNF-α, IFN-γ and TGF-β1) in the lung tissue of six mice in the PG–OJ group within 24 h, while the secondary figure shows the average changes within 2 h. (**D**) primarily illustrates the average tumor-growth-inhibition-rate changes within 24 h and 2 h for the same group of six mice in the PG–OJ group.

**Figure 4 molecules-29-00334-f004:**
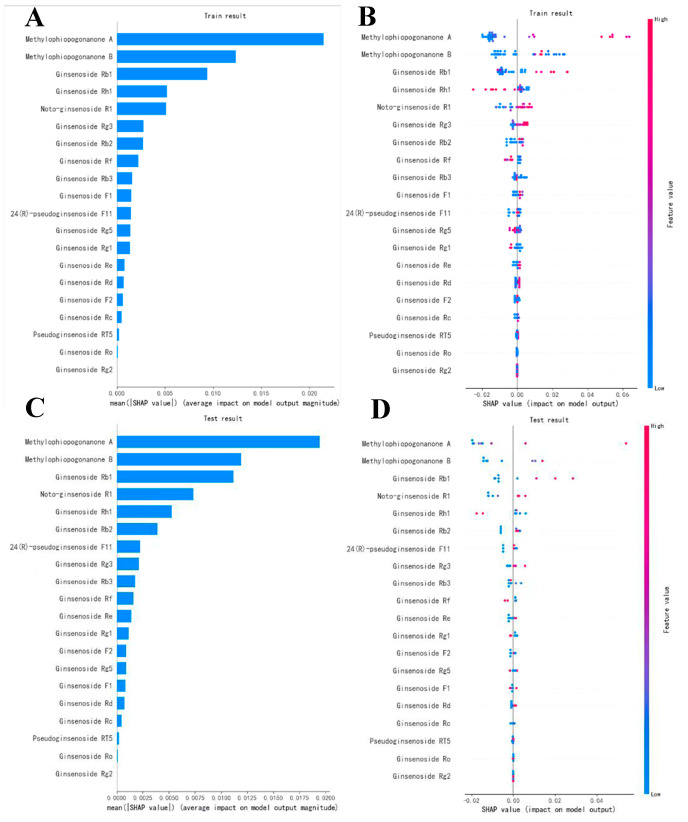
Feature-importance ranking of LightGBM model based on SHAP value. (**A**) Feature-importance ranking in the training set; (**B**) SHAP plot in the training set; (**C**) features-importance ranking in the test set; and (**D**) SHAP plot in the test set.

**Table 1 molecules-29-00334-t001:** The regression equation and linear range for 20 compounds from PG–OJ.

Compounds	Regression Equation	R^2^	Linear Range (ng/g)	LLOQ (ng/g)
Ginsenoside Rg_1_	y = −0.000025 + 0.00015 × x	0.9996	0.10–500	0.10
Ginsenoside Re	y = −0.0000039 + 0.0000078 × x	0.9995	0.10–500	0.10
Ginsenoside Ro	y = −0.00015 + 0.00012 × x	0.9996	0.25–500	0.25
Ginsenoside Rh_1_	y = 0.000060 + 0.00011 × x	0.9996	0.25–500	0.25
Ginsenoside F1	y = 0.0000053 + 0.00015 × x	0.9974	0.25–500	0.25
Ginsenoside F2	y = 0.000012 + 0.000052 × x	0.9993	0.50–500	0.50
Notoginsenoside R1	y = 0.0000060 + 0.00000026 × x	0.9990	0.005–500	0.005
Ginsenoside Rg_5_	y = −0.00042 + 0.000081 × x	0.9997	50–10,000	50.00
Ginsenoside Rg_2_	y = −0.0021 + 0.00065 × x	0.9998	0.25–500	0.25
Ginsenoside Rg_3_	y = 0.000012 + 0.00014 × x	0.9992	0.25–500	0.25
Ginsenoside Rf	y = −0.000062 + 0.000029 × x	0.9990	0.10–500	0.10
Ginsenoside Rb_1_	y = −0.000038 + 0.000068 × x	0.9997	0.10–500	0.10
Ginsenoside Rb_2_	y = −0.00011 + 0.000081 × x	0.9992	0.50–500	0.50
Ginsenoside Rb_3_	y = 0.00016 + 0.000037 × x	0.9993	0.50–500	0.50
Ginsenoside Rc	y = −0.00022 + 0.00014 × x	0.9998	0.50–500	0.50
Ginsenoside Rd	y = −0.00020 + 0.00016 × x	0.9997	0.25–500	0.25
Pseudoginsenoside RT_5_	y = 0.0052 + 0.00017 × x	0.9990	0.50–500	0.50
24(*R*)-pseudoginsenoside F11	y = 0.000044 + 0.00014 × x	0.9991	0.25–500	0.25
Methylophiopogon flavanone A	y = −0.00034 + 0.00021 × x	0.9997	0.25–500	0.25
Methylophiopogon flavanone B	y = −0.000089 + 0.00016 × x	0.9994	0.10–500	0.10

**Table 2 molecules-29-00334-t002:** The precision and accuracy of 20 compounds at low, medium, and high concentrations during inter-day and intra-day tests (*n* = 6).

Compounds	Nominal Concentration (ng/g)	Intra-Day		Inter-Day	
		Precision (RSD%)	Accuracy (RE%)	Precision (RSD%)	Accuracy (RE%)
Ginsenoside Rg_1_	5	3.67%	−0.58%	1.97%	2.68%
	50	3.71%	−0.31%	3.61%	1.03%
	500	8.28%	−4.01%	3.02%	−0.18%
Ginsenoside Re	5	5.13%	−0.10%	13.22%	10.71%
	50	12.09%	−14.43%	8.16%	−14.86%
	500	3.31%	−9.34%	11.62%	−4.12%
Ginsenoside Ro	5	7.50%	3.64%	12.85%	8.34%
	50	7.81%	4.45%	8.31%	2.31%
	500	12.85%	−1.01%	7.09%	−3.14%
Ginsenoside Rh_1_	5	4.94%	8.66%	5.90%	10.26%
	50	11.78%	−5.02%	8.44%	−1.37%
	500	11.26%	1.14%	10.67%	−0.60%
Ginsenoside F1	5	3.31%	3.81%	3.51%	7.55%
	50	4.44%	0.95%	3.88%	0.50%
	500	10.39%	−3.38%	5.70%	−1.42%
Ginsenoside F_2_	5	1.57%	0.05%	3.18%	−1.82%
	50	5.54%	1.29%	4.95%	2.65%
	500	10.39%	−2.04%	3.84%	1.04%
Notoginsenoside R1	5	10.28%	11.96%	6.94%	6.76%
	50	14.25%	9.24%	8.37%	10.25%
	500	5.78%	13.01%	4.87%	−9.83%
Ginsenoside Rg_5_	5	9.10%	−1.55%	14.73%	−5.73%
	50	6.14%	−3.75%	7.70%	−3.68%
	500	11.80%	−7.95%	4.94%	−0.09%
Ginsenoside Rg_2_	5	9.57%	11.69%	11.68%	12.11%
	50	8.30%	−3.65%	8.81%	−5.70%
	500	10.55%	9.49%	10.44%	6.54%
Ginsenoside Rg_3_	5	3.25%	6.73%	3.57%	5.42%
	50	3.24%	0.66%	4.73%	2.50%
	500	8.82%	−4.47%	2.50%	0.29%
Ginsenoside Rf	5	3.67%	−7.00%	8.84%	−4.19%
	50	7.33%	5.65%	9.05%	7.85%
	500	8.90%	−4.17%	4.00%	−3.07%
Ginsenoside Rb_1_	5	8.19%	5.68%	5.20%	10.27%
	50	9.10%	7.12%	7.53%	5.81%
	500	8.80%	−4.68%	6.24%	−2.28%
Ginsenoside Rb_2_	5	4.88%	0.84%	6.15%	8.20%
	50	8.44%	4.22%	8.43%	4.30%
	500	8.86%	−1.16%	5.00%	−2.66%
Ginsenoside Rb_3_	5	8.90%	−7.38%	11.52%	−7.37%
	50	7.04%	−2.91%	5.69%	0.04%
	500	13.20%	−5.55%	5.89%	2.28%
Ginsenoside Rc	5	4.14%	1.24%	9.36%	−5.75%
	50	10.62%	−8.09%	7.65%	−3.45%
	500	13.66%	−2.49%	11.64%	0.21%
Ginsenoside Rd	5	9.37%	5.41%	1.87%	11.06%
	50	3.58%	1.80%	5.40%	4.06%
	500	7.69%	−1.63%	3.51%	−0.32%
Pseudoginsenoside RT_5_	5	1.93%	0.94%	1.39%	2.94%
	50	3.52%	2.14%	8.05%	6.65%
	500	8.09%	−2.43%	3.09%	0.94%
24(*R*)-pseudoginsenoside F11	5	3.25%	4.81%	3.48%	4.74%
	50	3.22%	1.09%	4.88%	3.17%
	500	7.10%	−4.07%	1.89%	−0.19%
Methylophiopogon flavanone A	5	4.06%	9.29%	3.58%	9.21%
	50	3.60%	−1.90%	6.43%	0.56%
	500	8.39%	−3.36%	4.25%	−0.65%
Methylophiopogon flavanone B	5	3.59%	5.19%	6.61%	2.75%
	50	4.09%	−3.82%	6.37%	−1.21%
	500	8.78%	−2.83%	5.05%	0.27%

**Table 3 molecules-29-00334-t003:** Extraction recoveries and matrix effects for 20 compounds at high, medium, and low concentrations (*n* = 6).

Compounds	Nominal Concentration (ng/g)	Extraction Recovery		Matrix Effect	
		Mean ± SD (%)	RSD (%)	Mean ± SD (%)	RSD (%)
Ginsenoside Rg_1_	5	102.54 ± 5.35	5.22	91.20 ± 5.23	5.74
	50	99.25 ± 3.72	3.74	97.24 ± 6.01	6.18
	500	101.85 ± 1.70	1.67	105.54 ± 5.51	5.22
Ginsenoside Re	5	102.03 ± 4.98	4.88	105.94 ± 3.07	2.90
	50	99.28 ± 4.54	4.58	99.95 ± 5.12	5.12
	500	103.82 ± 3.06	2.95	101.40 ± 3.87	3.82
Ginsenoside Ro	5	109.20 ± 8.40	7.70	95.81 ± 3.90	4.07
	50	92.95 ± 2.63	2.83	90.57 ± 5.89	6.50
	500	105.26 ± 2.48	2.36	100.70 ± 7.99	7.93
Ginsenoside Rh_1_	5	100.56 ± 6.15	6.12	107.55 ± 0.68	0.63
	50	98.56 ± 3.36	3.41	95.99 ± 4.47	4.65
	500	104.53 ± 1.30	1.24	103.87 ± 3.29	3.17
Ginsenoside F1	5	100.83 ± 4.76	4.72	103.12 ± 0.50	0.49
	50	99.35 ± 3.47	3.49	94.52 ± 4.62	4.88
	500	103.32 ± 0.61	0.59	101.06 ± 5.51	5.15
Ginsenoside F_2_	5	101.19 ± 4.58	4.52	103.50 ± 2.67	2.58
	50	99.81 ± 8.81	8.83	102.04 ± 6.75	6.61
	500	102.48 ± 0.47	0.46	101.14 ± 5.17	5.11
Notoginsenoside R1	5	106.68 ± 5.63	5.28	108.68 ± 1.39	1.28
	50	105.11 ± 6.07	5.77	104.17 ± 4.13	3.96
	500	90.53 ± 4.55	5.03	104.16 ± 2.47	2.37
Ginsenoside Rg_5_	5	88.79 ± 9.64	10.86	94.04 ± 4.37	4.65
	50	102.14 ± 9.37	9.17	91.92 ± 9.32	10.14
	500	105.76 ± 4.70	4.44	88.85 ± 8.56	9.63
Ginsenoside Rg_2_	5	101.42 ± 6.48	6.39	87.43 ± 3.31	3.79
	50	102.51 ± 2.42	2.36	96.56 ± 5.11	5.29
	500	101.75 ± 6.75	6.64	100.94 ± 3.36	3.33
Ginsenoside Rg_3_	5	103.19 ± 5.52	5.35	100.84 ± 9.24	9.16
	50	101.82 ± 1.71	1.68	99.60 ± 4.06	4.08
	500	101.00 ± 2.96	2.93	105.45 ± 5.09	4.82
Ginsenoside Rf	5	101.09 ± 1.45	1.43	106.77 ± 1.76	1.65
	50	96.94 ± 1.10	1.13	94.08 ± 4.43	4.71
	500	93.95 ± 0.50	0.53	95.49 ± 8.33	8.73
Ginsenoside Rb_1_	5	97.98 ± 1.49	1.53	104.16 ± 10.06	9.66
	50	94.94 ± 3.94	4.15	91.29 ± 8.92	9.78
	500	94.63 ± 1.60	1.69	95.02 ± 3.24	3.41
Ginsenoside Rb_2_	5	106.83 ± 10.37	9.71	91.96 ± 3.36	3.65
	50	90.20 ± 0.95	1.06	86.80 ± 3.97	4.57
	500	99.93 ± 0.21	0.21	90.85 ± 1.33	1.46
Ginsenoside Rb_3_	5	100.26 ± 0.12	0.12	103.25 ± 1.03	1.00
	50	97.90 ± 2.76	2.82	91.06 ± 9.65	9.60
	500	105.68 ± 4.45	4.21	101.07 ± 0.79	0.78
Ginsenoside Rc	5	106.00 ± 9.49	8.96	103.24 ± 2.19	2.12
	50	105.32 ± 5.84	5.54	97.35 ± 2.63	2.70
	500	103.82 ± 3.06	2.95	89.15 ± 3.08	3.45
Ginsenoside Rd	5	94.57 ± 2.18	2.31	101.35 ± 1.99	1.97
	50	91.97 ± 9.20	10.01	90.50 ± 11.17	12.34
	500	87.71 ± 1.24	1.42	103.30 ± 3.12	3.02
Pseudoginsenoside RT_5_	5	105.81 ± 11.39	10.76	88.10 ± 1.22	1.38
	50	99.32 ± 0.46	0.46	97.33 ± 2.49	2.56
	500	105.63 ± 7.06	6.68	92.97 ± 8.64	9.29
24(*R*)-pseudoginsenoside F11	5	100.44 ± 4.78	4.66	104.37 ± 2.60	2.49
	50	100.94 ± 2.78	2.75	98.85 ± 5.13	5.19
	500	96.92 ± 5.55	5.72	99.30 ± 6.58	6.63
Methylophiopogon flavanone A	5	101.56 ± 1.45	1.43	91.33 ± 1.39	1.52
	50	105.05 ± 7.10	6.76	99.85 ± 1.44	1.44
	500	88.99 ± 0.91	1.02	93.11 ± 3.34	3.59
Methylophiopogon flavanone B	5	99.45 ± 3.02	3.04	98.99 ± 0.95	0.96
	50	105.41 ± 22.49	2.36	103.36 ± 5.31	5.14
	500	97.94 ± 3.04	3.11	87.71 ± 6.06	6.91

**Table 4 molecules-29-00334-t004:** Stability of 20 compounds in lung tissue under different conditions (*n* = 6).

Compounds	Nominal Concentration (ng/g)	Short Term		Three Freeze–Thaw Cycles		Long Term	
		RSD (%)	RE (%)	RSD (%)	RE (%)	RSD (%)	RE (%)
Ginsenoside Rg_1_	5	3.74%	0.49%	10.56%	11.24%	6.12%	−1.70%
	50	3.94%	1.34%	7.41%	13.01%	4.32%	17.42%
	500	10.80%	0.20%	5.53%	14.49%	5.64%	13.31%
Ginsenoside Re	5	11.05%	4.81%	10.92%	−1.98%	7.39%	−19.79%
	50	12.72%	−13.15%	0.62%	10.81%	2.00%	14.21%
	500	11.79%	6.17%	16.02%	11.65%	18.75%	13.94%
Ginsenoside Ro	5	8.54%	8.62%	11.19%	16.33%	10.47%	8.22%
	50	7.84%	6.89%	10.41%	10.39%	12.03%	13.22%
	500	12.82%	0.28%	11.66%	15.72%	10.51%	11.59%
Ginsenoside Rh_1_	5	5.55%	11.68%	6.01%	15.36%	4.53%	−1.41%
	50	12.06%	−3.24%	11.28%	18.32%	15.41%	18.76%
	500	8.80%	−0.38%	2.83%	10.57%	5.48%	17.88%
Ginsenoside F1	5	4.17%	6.91%	9.85%	18.33%	10.67%	−4.70%
	50	3.83%	2.39%	9.37%	16.72%	7.36%	19.31%
	500	11.39%	0.31%	6.12%	9.29%	9.18%	12.65%
Ginsenoside F2	5	2.23%	−1.67%	7.94%	7.15%	4.74%	−2.19%
	50	5.94%	4.33%	6.64%	13.85%	7.13%	15.28%
	500	10.74%	0.36%	3.29%	−1.50%	9.81%	3.40%
Notoginsenoside R1	5	10.26%	11.06%	8.95%	14.20%	2.96%	4.89%
	50	8.53%	6.89%	2.12%	6.22%	2.71%	8.60%
	500	11.57%	−3.93%	16.60%	−5.12%	13.87%	−7.16%
Ginsenoside Rg_5_	5	12.61%	−9.35%	19.09%	19.21%	7.46%	15.87%
	50	8.07%	−7.55%	9.95%	9.55%	10.76%	−6.36%
	500	10.82%	5.42%	5.80%	16.06%	6.71%	14.05%
Ginsenoside Rg_2_	5	3.25%	11.46%	16.98%	−18.16%	5.66%	−2.28%
	50	7.42%	0.76%	16.66%	2.30%	16.66%	2.30%
	500	14.04%	2.58%	4.29%	−3.34%	18.65%	−17.08%
Ginsenoside Rg_3_	5	2.48%	5.07%	11.40%	17.95%	4.71%	15.74%
	50	3.60%	2.96%	3.46%	0.71%	3.46%	0.71%
	500	11.75%	−0.04%	2.95%	17.81%	3.75%	18.93%
Ginsenoside Rf	5	3.27%	−7.44%	19.17%	17.80%	13.23%	−1.25%
	50	6.34%	11.93%	9.94%	18.33%	9.94%	18.33%
	500	11.26%	2.59%	5.47%	9.47%	5.14%	7.20%
Ginsenoside Rb_1_	5	7.45%	3.93%	17.12%	3.07%	5.97%	−14.73%
	50	7.74%	12.32%	7.54%	17.02%	7.43%	16.77%
	500	11.36%	2.98%	16.29%	13.24%	2.37%	19.79%
Ginsenoside Rb_2_	5	7.03%	5.17%	17.85%	12.19%	2.61%	19.17%
	50	8.15%	8.58%	7.01%	18.46%	6.44%	18.92%
	500	10.19%	0.07%	4.70%	17.45%	7.06%	19.95%
Ginsenoside Rb_3_	5	7.80%	−10.06%	5.93%	12.30%	2.66%	−6.18%
	50	8.50%	−1.19%	6.94%	−7.33%	5.26%	−5.61%
	500	10.68%	4.01%	2.62%	14.85%	8.13%	18.86%
Ginsenoside Rc	5	7.90%	−5.12%	14.41%	−9.09%	4.15%	−13.72%
	50	11.44%	−9.49%	7.40%	1.40%	7.40%	1.40%
	500	18.90%	−0.11%	9.88%	−1.45%	8.64%	0.82%
Ginsenoside Rd	5	7.45%	7.45%	8.21%	12.09%	4.08%	4.40%
	50	4.68%	5.45%	7.52%	−12.02%	8.34%	−10.53%
	500	8.84%	−0.73%	2.55%	14.57%	1.57%	15.40%
Pseudoginsenoside RT_5_	5	2.11%	1.84%	13.25%	8.07%	2.11%	2.57%
	50	6.97%	8.24%	5.46%	2.66%	17.16%	19.04%
	500	8.85%	−1.16%	2.06%	12.89%	2.37%	13.24%
24(*R*)-pseudoginsenoside F_11_	5	2.27%	3.05%	11.45%	9.60%	4.73%	7.63%
	50	3.64%	3.81%	5.78%	8.32%	5.78%	8.32%
	500	10.45%	0.06%	2.31%	17.38%	2.26%	17.29%
Methyl ophiopogon flavanone A	5	3.20%	8.50%	18.33%	−10.97%	3.26%	−17.92%
	50	6.69%	0.25%	6.76%	−14.39%	7.63%	−13.20%
	500	10.48%	−0.92%	8.68%	−19.99%	9.02%	−19.61%
Methyl ophiopogon flavanone B	5	3.11%	4.61%	18.61%	−16.54%	8.96%	−17.22%
	50	6.34%	−2.69%	8.47%	15.54%	7.37%	12.48%
	500	8.83%	1.73%	17.70%	13.69%	18.75%	12.19%

**Table 5 molecules-29-00334-t005:** Pharmacokinetic parameters of isolated components after intragastric administration (mean ± SD, *n* = 6).

Components	C_max_/ng/g	T_max_/min	AUC_(0–∞)_/ng/g·min	AUC_(0–t)_/ng/g·min	MRT_(0–t)_/min	T_1/2_/min
Ginsenoside Rg_1_	50.24 ± 6.67	15.00 ± 0.00	2345.90 ± 212.68	2295.03 ± 213.32	135.88 ± 4.70	98.12 ± 2.74
Ginsenoside Re	132.49 ± 32.88	5.00 ± 0.00	3933.20 ± 224.66	3815.68 ± 233.30	93.98 ± 3.79	105.69 ± 4.36
Ginsenoside Ro	11.95 ± 0.88	12.50 ± 2.89	1415.87 ± 57.44	1043.18 ± 64.96	184.92 ± 4.08	252.46 ± 11.88
Ginsenoside Rh_1_	8.98 ± 0.20	12.50 ± 5.00	1724.14 ± 411.48	1303.44 ± 70.16	190.78 ± 1.61	418.21 ± 293.98
Ginsenoside F1	4.12 ± 0.45	240.00 ± 0.00	648.77 ± 60.38	640.00 ± 61.94	197.55 ± 1.51	134.21 ± 14.44
Ginsenoside F2	134.49 ± 1.58	60.00 ± 0.00	24,395.33 ± 613.30	21,872.62 ± 480.93	184.97 ± 1.87	349.67 ± 16.15
Ginsenoside Rg_5_	7967.81 ± 441.77	240.00 ± 0.00	1,372,202.28 ± 48,963.33	1,342,117.43 ± 45,675.73	183.91 ± 2.00	126.76 ± 9.45
Ginsenoside Rg_2_	71.11 ± 6.20	15.00 ± 0.00	16,012.21 ± 284.14	13,822.64 ± 316.65	187.28 ± 1.45	214.68 ± 6.15
Ginsenoside Rg_3_	4.75 ± 0.12	5.00 ± 0.00	14,133.43 ± 3134.88	1879.87 ± 53.24	230.11 ± 2.69	2381.66 ± 562.33
Ginsenoside Rf	47.90 ± 6.96	15.00 ± 0.00	3678.93 ± 167.94	3656.82 ± 167.79	159.91 ± 1.98	76.57 ± 1.04
Ginsenoside Rb_1_	440.68 ± 15.67	720.00 ± 0.00	283,948.70 ± 17,261.40	227,245.91 ± 5460.47	687.50 ± 1.03	1974.59 ± 188.71
Ginsenoside Rb_2_	7.20 ± 0.96	720.00 ± 0.00	6076.50 ± 664.68	3935.63 ± 598.40	712.61 ± 2.29	1191.02 ± 262.76
Ginsenoside Rb_3_	5.92 ± 0.45	720.00 ± 0.00	16,689.44 ± 2585.97	4049.29 ± 622.84	827.16 ± 190.05	3995.02 ± 516.27
Ginsenoside Rc	42.95 ± 2.11	720.00 ± 0.00	25,129.88 ± 1611.92	23,180.66 ± 703.80	698.08 ± 7.43	444.67 ± 101.01
Ginsenoside Rd	82.95 ± 2.11	720.00 ± 0.00	56,296.57 ± 6663.99	46,163.94 ± 1033.00	667.90 ± 1.77	970.90 ± 220.59
Pseudoginsenoside RT_5_	5.11 ± 0.88	15.00 ± 0.00	2160.58 ± 307.18	890.71 ± 40.23	224.26 ± 5.19	634.34 ± 141.79
Notoginsenoside R1	7.16 ± 1.65	7.50 ± 3.54	1703.60 ± 109.17	1636.22 ± 104.54	207.16 ± 5.18	74.38 ± 1.47
24(*R*)-pseudoginsenoside F11	47.49 ± 6.94	15.00 ± 0.00	5604.97 ± 147.55	4660.97 ± 205.76	184.94 ± 2.87	227.16 ± 12.27
Methylophiopogonanone A	5.70 ± 0.88	150.00 ± 0.00	5425.63 ± 1564.22	1362.39 ± 36.13	219.18 ± 4.79	1219.25 ± 362.74
Methylophiopogonanone B	20.61 ± 2.38	150.00 ± 0.00	3466.12 ± 558.35	2331.08 ± 125.83	184.20 ± 10.94	390.53 ± 100.42

**Table 6 molecules-29-00334-t006:** Results of efficacy index in each group of mice (mean ± SD, *n* = 6).

Group	TNF-α (pg/g)	IFN-γ (pg/g)	TGF-β1 (pg/g)	Tumor Weight (mg)
C	6.25 ± 1.38 ***	541.81 ± 37.39 ***	626.67 ± 25.88 ***	----
M	2.30 ± 0.38	53.31 ± 7.41	316.33 ± 88.55	1282.09 ± 181.31
0.083 h	5.63 ± 0.66 **	500.92 ± 199.42 ***	395.33 ± 79.28	519.22 ± 72.02 ***
0.167 h	2.79 ± 1.38	697.58 ± 221.38 ***	492.67 ± 124.69	513.57 ± 24.41 ***
0.25 h	3.53 ± 0.97	272.42 ± 71.31	460.83 ± 75.59	571.74 ± 64.86 ***
0.5 h	3.84 ± 1.11	206.23 ± 66.60	720.00 ± 23.53 ***	483.84 ± 89.12 ***
0.75 h	5.01 ± 1.31 *	167.15 ± 88.39	766.22 ± 112.59 ***	499.70 ± 40.04 ***
1 h	4.73 ± 0.75 *	79.58 ± 12.22	511.50 ± 98.52 *	513.44 ± 16.94 ***
2 h	5.64 ± 1.47 **	140.73 ± 18.45	761.17 ± 109.49 ***	577.87 ± 40.68 ***
3 h	6.97 ± 1.44 ***	522.69 ± 116.26 ***	975.00 ± 104.86 ***	614.60 ± 39.46 ***
4 h	3.35 ± 1.43	132.55 ± 53.12	554.67 ± 20.40 **	611.46 ± 12.89 ***
6 h	2.86 ± 1.01	82.68 ± 11.38	510.67 ± 104.52 *	578.167 ± 19.38 ***
8 h	2.42 ± 0.45	71.11 ± 7.73	467.11 ± 71.36	608.35 ± 27.81 ***
12 h	2.56 ± 0.67	72.67 ± 8.90	472.71 ± 51.56	660.64 ± 42.43 ***
24 h	2.65 ± 0.97	75.15 ± 12.45	487.83 ± 64.64	626.68 ± 23.61 ***

Note: Compared with the model group, * *p* < 0.05, ** *p* < 0.01, *** *p* < 0.001.

## Data Availability

All data generated or analyzed during this study are included in this manuscript and the Appendix A.

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
