# Peer review of "Using Pharmacokinetic–Pharmacodynamic Modeling to Study the Main Active Substances of the Anticancer Effect in Mice from Panax ginseng–Ophiopogon japonicus"

_molecules, 2024, doi:10.3390/molecules29020334_

Round 1
Reviewer 1 Report
Comments and Suggestions for Authors
Authors reported a study of PK-PD modeling to study the "material basis" of anticancer effect of a combined preparation of PG-OJ. The study provides insight in the PK-PD correlation of the treatment, and has the merit for publication. However, there are some issues which authors should attempt to address. The comments are given below.
Title: Please rephrase "material basis" - it is not clear as to what "materials" do you mean? Also, please include the animal model (in mice...) to ensure it is clear from the title that this is not a clinical / human study.
Line 37-38: Replace "external forces" and "activating the body immune system" with more proper English phrases.
Line 49-50: The comparability ofP. ginseng and O. japonicus (PG-PJ) has led to the development of... -- please provide further explanation on this subject. What do you mean with "comparability"? Any reference?
Line 54: What is "material basis"... and how or why this was not covered by previous studies?
Line 64-65: check for typo (machine), and inverted comma.
Line 74-75: explain further what do you mean with "chemical components of different time points in lung tissues..." - Are you referring to all the proteins or mediators in any lung tissues of mice? Under what conditions?
Results
Figure 2 and Figure 3: Please provide descriptions in the caption regarding the experimental method in brief, animal model, number of mice, and the grouping. Were the values represent means of how many specimens, and any error bars presented? Also please provide legend for the inlet (inserted) time course profile.
Discussion:
First paragraph: I'd suggest authors to include a brief explanation of the choice of 20 targeted compounds as shown in their results. Provide the relevant references also for the detection and importance of these compounds.
Line 261-292:
The discussion within this paragraph can be improved by providing more insights from the correlation between PK (of the compounds) and PD (the anti-tumor effect). As it is, there is a lack of comparison between the groups, and more in-depth discussion on this. Are there any differences between the treated and untreated groups in terms of tumor growth/reduction, and were these supported by or correlated with the PK of PG-OJ?
Additionally, authors should include limitations of the study.
Comments on the Quality of English Language
Some phrases, e.g. "external forces", sound more like direct translation from Chinese. Suggest to rephrase for clarity.
Author Response
Question 1: Title: Please rephrase "material basis" - it is not clear as to what "materials" do you mean? Also, please include the animal model (in mice...) to ensure it is clear from the title that this is not a clinical / human study.
Response: We are so sorry for the confusing understanding caused by our misstatement. We have modified "material basis" to the “Main Active Substances”. We have modified it to the following sentence. Title: Pharmacokinetic-Pharmacodynamic Modeling to Study the Main Active Substances of Anticancer Effect in Mice from Panax ginseng - Ophiopogon japonicus.
Question 2: Line 37-38: Replace "external forces" and "activating the body immune system" with more proper English phrases.
Response: Thanks for your advice. I have modified this sentence to “In the field of cancer treatment, the shift from relying on external interventions like surgery, chemotherapy, or radiotherapy to activating the human immune system is crucial.”
Question 3: Line 49-50: The comparability of P. ginseng and O. japonicus (PG-PJ) has led to the development of... -- please provide further explanation on this subject. What do you mean with "comparability"? Any reference?
Response: We thank the reviewer for pointing out this issue. “The compatibility of P. ginseng and O. japonicas (PG-OJ) has led to the development of an injectable formulation, known as Shenmai Injection.” To further clarify this subject, we provide the following explanation and relevant references. “Comparability” refers to the compatibility or synergy between Panax ginseng and Ophiopogon japonicus, which is the basis for their combined use in traditional Chinese medicine. This concept is well-documented in numerous studies, indicating that the two herbs can be used together effectively for therapeutic purposes.
Here are some relevant references to support this explanation:
- Zhang S, Sun H, Wang C, et al. Comparative analysis of active ingredients and effects of the combination of Panax ginseng and Ophiopogon japonicus at different proportions on chemotherapy-induced myelosuppression mouse. Food Funct. 2019, 10(3): 1563-1570.
- Shi, L, Xie, Y, Liao, X, et al. Shenmai injection as an adjuvant treatment for chronic cor pulmonale heart failure: a systematic review and meta-analysis of randomized controlled trials. BMC Complement Altern Med. 2015, 15:418.
- Xian, S, Yang, Z, Lee, J, et al. A randomized, double-blind, multicenter, placebo-controlled clinical study on the efficacy and safety of Shenmai injection in patients with chronic heart failure. J Ethnopharmacol. 2016, 186:136–42.
- Liang Y, Zhou Y, Zhang J, et al. Pharmacokinetic compatibility of ginsenosides and Schisandra Lignans in Shengmai-san: from the perspective of p-glycoprotein. PloS one 2014, 9(6): e98717.
Question 4: Line 54: What is "material basis"... and how or why this was not covered by previous studies?
Response: Material basis refers to the active substances in a drug that contribute to its therapeutic effects. In this study, it refers to the biologically active compounds in the PG-OJ herbal combination that exhibit antitumor properties. Previous studies have focused on the efficacy or chemical composition of PG-OJ, but have not explored the correlation between these components and the therapeutic effects. Therefore, a systematic study on the material basis of PG-OJ's antitumor effects is lacking. This is the gap that our study aims to fill. Our research will provide a deeper understanding of the specific components responsible for the antitumor activity of PG-OJ, contributing to the advancement of herbal medicine in cancer treatment.
Question 5: Line 64-65: check for typo (machine), and inverted comma.
Response: Thank you very much for your suggestion. I have revised it in the revised manuscript.
Question 6: Line 74-75: explain further what do you mean with "chemical components of different time points in lung tissues..." - Are you referring to all the proteins or mediators in any lung tissues of mice? Under what conditions?
Response: Thank you very much for your suggestion. We are referring to the metabolite prototypes in lung tissues of mice at different time points. We use UPLC-MS/MS technology to comprehensively analyze and characterize these chemical compounds under various conditions. This helps us to understand the dynamic changes in metabolic pathways and potential biomarkers associated with the lungs of mice over time.
Results
Question 7: Figure 2 and Figure 3: Please provide descriptions in the caption regarding the experimental method in brief, animal model, number of mice, and the grouping. Were the values represent means of how many specimens, and any error bars presented? Also please provide legend for the inlet (inserted) time course profile.
Response: We thank the reviewer for pointing out this issue. We have modified it to the following sentence. “Figure 2. The profiles of mean lung tissues concentration − time of twenty target constituents after oral administration of PG-OJ extract to tumor-bearing mice. The main figure displays the average content changes of components in the lung tissue of six mice in the PG-OJ group within 24 hours, while the secondary figure shows the average content changes within 2 hours. The error lines represent the mean ± SD.” “Figure 3.Pharmacodynamic indicator changes in tumor-bearing mice after oral administration of PG-OJ extract at different time points. Main figures A-C present the average repair rate changes of three cytokines (TNF-α, IFN-γ and TGF-β1) in the lung tissue of six mice in the PG-OJ group within 24 hours, while the secondary figure shows the average changes within 2 hours. Figure D primarily illustrates the average tumor growth inhibition rate changes within 24 hours and 2 hours for the same group of six mice in the PG-OJ group.”
Discussion:
Question 8: First paragraph: I'd suggest authors to include a brief explanation of the choice of 20 targeted compounds as shown in their results. Provide the relevant references also for the detection and importance of these compounds.
Response: Thank you very much for your suggestion. We have added a brief explanation of the selection of 20 targeted compounds in section 2.1.1 of the manuscript. We have also provided relevant references regarding the detection and significance of these compounds. Not all of these compounds were detected in lung tissue samples from tumor-bearing mice due to different bioavailability profiles. We carefully considered compounds with distinct pharmacological properties and therapeutic potential, as well as those that are relevant to our research question. Based on our UPLC-MS/MS analysis, we successfully identified and selected these 20 prototype components for further pharmacokinetic study. We have also provided relevant references to support the detection and importance of these compounds. We hope this revised explanation addresses your concern.
Reference:
- Xu, F.Y.; Shang, W.Q.; Yu, J.J.; Sun, Q.; Li, M.Q.; Sun J.S. The antitumor activity study of ginsenosides and metabolites in lung cancer cell. J. Transl. Res. 2016, 8, 1708.
- Jung, D.H.; Nahar, J.; Mathiyalagan, R.; Rupa, E.J.; Ramadhania, Z.M.; Han, Y.; Yang, Chun.; Kang, S.C. A focused review on molecular signalling mechanisms of ginsenosides anti-Lung cancer and anti-inflammatory activities. Anti-Cancer Agents Med. Chem. 2023, 23, 3-14.
Question 9: Line 261-292:
The discussion within this paragraph can be improved by providing more insights from the correlation between PK (of the compounds) and PD (the anti-tumor effect). As it is, there is a lack of comparison between the groups, and more in-depth discussion on this. Are there any differences between the treated and untreated groups in terms of tumor growth/reduction, and were these supported by or correlated with the PK of PG-OJ? Additionally, authors should include limitations of the study.
Response: Thank you for your valuable feedback. We agree that the discussion on the correlation between PK and PD could be enhanced. In response to your suggestion, we have added more comparisons between treated and untreated groups, but the difference in efficacy and its correlation with PK still need further experimental exploration. The revised paragraph will strengthen the overall manuscript. In the revised paper, we will also include the limitation section of the study.
Question 10: Comments on the Quality of English Language
Some phrases, e.g. "external forces", sound more like direct translation from Chinese. Suggest to rephrase for clarity.
Response: We apologize for the poor language of our manuscript. We have now worked on both language and readability and have also involved native English speakers for language corrections.

Reviewer 2 Report
Comments and Suggestions for Authors
This study established a specific and sensitive LC-MS/MS method for simultaneous quantification of 18 saponins and two flavanones in biological samples and was successfully applied to analyze the pharmacokinetic characteristics of compounds in lung cancer mice by way of continuous gavage with PG-OJ extract for twenty-one days. The improvement rates of TNF-α, TGF-β1, and IFN-γ indicate that PG-OJ can effectively enhance the anti-tumor immune effect. Based on the LightGBM algorithm, the PK-PD pattern analysis revealed the importance ranking of 20 key components in lung cancer mice, and identified four key pharmacodynamic substance component groups.
The methods used in this study are sound and the results are clearly presented. It would be nice if the authors explain and discuss more about the dose of the extract used in the experiment and also the effective dose or formula which are previously reported.
Please check line 55 and 58 "RG"
Author Response
Reviewer 2:
Question 1: The methods used in this study are sound and the results are clearly presented. It would be nice if the authors explain and discuss more about the dose of the extract used in the experiment and also the effective dose or formula which are previously reported.
Response: Thank you for your suggestion. Regarding the dosage issue, the extract dosage used in our experiment was determined based on previous experimental experience and literature reports. Our previous research has demonstrated that PG-OJ extracts exhibit excellent anti-lung cancer therapeutic effects at a four-fold clinical equivalent dose. In this study, we will further validate the efficacy of this dose and explore its pharmacokinetic characteristics. In the upcoming revision, we will explain the reasons for selecting the dosage. However, since the existing literature mainly focused on the combination of PG-OJ and chemotherapy agents, we were unable to find an effective dose for direct reference. In subsequent studies, we will intensify our exploration of dosage aspects and strive to identify more precise dosage ranges. We appreciate your valuable suggestions and will strive to improve the paper accordingly.
Question 2: Please check line 55 and 58 "RG"
Response: Thank you very much for your suggestion. I have revised it in the revised manuscript.
